# Respiratory Inductance Plethysmography to Assess Fatigability during Repetitive Work

**DOI:** 10.3390/s22114247

**Published:** 2022-06-02

**Authors:** Luís Silva, Mariana Dias, Duarte Folgado, Maria Nunes, Praneeth Namburi, Brian Anthony, Diogo Carvalho, Miguel Carvalho, Elazer Edelman, Hugo Gamboa

**Affiliations:** 1Laboratório de Instrumentação, Engenharia Biomédica e Física da Radiação (LIBPhys-UNL), Departamento de Física, Faculdade de Ciências e Tecnologia, Universidade Nova de Lisboa, 2829-516 Caparica, Portugal; marianaagdias97@gmail.com (M.D.); duarte.folgado@fraunhofer.pt (D.F.); hgamboa@fct.unl.pt (H.G.); 2Associação Fraunhofer Portugal Research, Rua Alfredo Allen 455/461, 4200-135 Porto, Portugal; maria.nunes@aicos.fraunhofer.pt; 3Institute for Medical Engineering and Science, Massachusetts Institute of Technology, Cambridge, MA 02139, USA; praneeth.namburi@gmail.com (P.N.); banthony@mit.edu (B.A.); ere@mit.edu (E.E.); 4MIT.nano Immersion Lab, Massachusetts Institute of Technology, Cambridge, MA 02139, USA; 5Device Realization Laboratory, Department of Mechanical Engineering, Massachusetts Institute of Technology, Cambridge, MA 02139, USA; 6Faculty of Medicine, Rīga Stradiņš University, 16 Dzirciema iela, LV-1007 Rīga, Latvia; diogocarvalho98@hotmail.com; 7Campus de Azurém, Minho University, 4800-058 Guimarães, Portugal; migcar@det.uminho.pt; 8Brigham and Women’s Hospital, Cardiovascular Division, 75 Francis Street, Boston, MA 02115, USA

**Keywords:** RIP, EMG, fatigue, occupational risk, work, industry 4.0, operator 4.0

## Abstract

Cumulative fatigue during repetitive work is associated with occupational risk and productivity reduction. Usually, subjective measures or muscle activity are used for a cumulative evaluation; however, Industry 4.0 wearables allow overcoming the challenges observed in those methods. Thus, the aim of this study is to analyze alterations in respiratory inductance plethysmography (RIP) to measure the asynchrony between thorax and abdomen walls during repetitive work and its relationship with local fatigue. A total of 22 healthy participants (age: 27.0 ± 8.3 yrs; height: 1.72 ± 0.09 m; mass: 63.4 ± 12.9 kg) were recruited to perform a task that includes grabbing, moving, and placing a box in an upper and lower shelf. This task was repeated for 10 min in three trials with a fatigue protocol between them. Significant main effects were found from Baseline trial to the Fatigue trials (*p* < 0.001) for both RIP correlation and phase synchrony. Similar results were found for the activation amplitude of agonist muscle (*p* < 0.001), and to the muscle acting mainly as a joint stabilizer (*p* < 0.001). The latter showed a significant effect in predicting both RIP correlation and phase synchronization. Both RIP correlation and phase synchronization can be used for an overall fatigue assessment during repetitive work.

## 1. Introduction

Fatigue-inducing repetitive work is a widely known risk factor for occupational-related upper-limb musculoskeletal disorders [1,2,3], often requiring a consolidation of mental and physical demands [4]. Fatigue may interfere with the work itself by decreasing efficiency, and if it is excessive and extended, it may alter the worker’s mental and physical conditions, giving rise to illness or worsen a preexisting condition [5,6], even in tasks requiring high vigilance but low neuromuscular work [4]. This is a concern if we take into consideration that people worked at their head job on average 37.1 h a week in Europe [7] and 34.9 h in the United States [8]. Additionally, approximately one-third of the global workforce works more than 48 h per week [9]. Furthermore, work-related fatigue can bring consequences beyond the occupational risk and work-related illness, it fosters the occurrence of mistakes due to tiredness, which has severe implications in productivity and promotes absenteeism [10,11].

To avoid workers’ fatigue, ergonomic interventions such as a job rotation plan and breaks have been suggested, but the former do not demonstrate a significant effect on fatigue at lower exertion tasks [3]. On the other hand, the implementation of breaks requires fixed work/rest ratio or workload variation based on postures and movement.

The assessment of occupational risk can be performed with several methods classified according to three main approaches: self-reported, observational, and directly measured. Self-reported methods rely on questionnaires based on physical demand perception. However, they require interruptions of the worker’s activity and are prone to self-reporting bias arising from external perceived factors (e.g., behavioral and physiological factors). Observational methods are conducted by an external observer watching a worker performing his job and then filling appropriate standard instruments and ergonomic risk assessment tools (e.g., [12,13,14]). Direct methods use sensors that give physiological, movement and position, and/or environmental information (e.g., [15,16,17]). The latter has been gaining attention over the last years with the technological developments in wearables and data analytics surrounding the industry Operator 4.0 [18].

Regarding fatigue evaluation, empirical and theoretical models have been introduced, but they require individual calibration through fitting experimental data for a specific task [19]. Fatigue evaluation using wearables has recently attracted increasing attention as a promising approach to overcome the challenges associated with previous methods [20,21]. Wearables allow ubiquitous data collection in the workplace and combined with data analytics can provide useful information to enable healthy operators 4.0 [22,23]. In this sense, fatigue at low exertion load is usually evaluated by electromyography (EMG) [24]; nevertheless, also respiration biomarkers have been proposed based on the rib cage and abdominal asynchrony [25] and hyperventilation hypotheses [26]. These biomarkers can give information on overall tiredness rather than local, or even the effects of local fatigue on general fatigability and tiredness. We further provide an overview of such methods and identify their limitations.

EMG fatigue manifestation is observed by a decrease in signal median frequency and an increase in amplitude observed over time [24,27]. Compared to activities performed at high-level intensity, the quantification of fatigue due to low-level force is not so evident. However, fatigue can be used as a biomarker for cumulative exposure to repetitive work [28]. In amplitude analysis, standardized voluntary contractions have frequently been used, in which maximal voluntary contraction (MVC) tends to be the preferred reference choice (e.g., [29,30]). In the review of de Looze et al. [24], the MVC values that quantified a low load characterizing repetitive work ranged between 3 up to the cut-point of 20% of MVC in trapezius muscle, in which fatigue was found to be prevalent between 15 and 20% of MVC with a task duration varying between 1 h up to 9.5 h. Due to the low exertion during long periods associated with work unlike sports, amplitude probability distribution function [31] is commonly used to quantify local muscular workload [32,33]. The median frequency, when showing alterations, decreases ranging between 2 and 15% per hour [34]. However, in workplaces where the effort is almost at a lower level, this indicator could be insufficient or dependent on the posture fluctuations [35]. The EMG, regardless its relevance, presents several drawbacks, such as: (1) can only provide local muscle load, lacking on overall physical load [36]; and (2) it requires MVC procedure, which is difficult to adapt to the non-laboratory environment of industrial assembly lines settings.

To overcome overall physical load, the relationship between reserve oxygen consumption and heart rate (HR) has been a suitable option [37,38] since the former is the product between the arteriovenous difference and cardiac output (the product between HR and stroke volume). An indicator that works as an index of the intensity of physical work is the relative HR (e.g., [39,40]), i.e., the ratio of the difference between work and rest HR with the reserve HR (maximum HR–rest HR). From relative HR, the maximum acceptable work time (MAWT) balance between cardiovascular load and physical workload can be determined. Both relative HR and MAWT indicators were associated with the percentage of maximal oxygen consumption and work-related oxygen consumption [41].

Considering the previous paragraph, it is reasonable to wonder whether MAWT can be associated with other respiratory concepts such as asynchrony and hyperventilation theories. The former states that, during repetitive work, there is an asynchrony between the work rhythm and a worker’s internal physiological rhythms which is a source of both motor and cognitive stress [25]. The alterations in respiratory function can be either derived from intense or continued physical exertion, or from disease. Thoraco-abdominal (a)synchrony can be used to determine respiratory disorders and/or respiratory muscle dysfunctions associated with respiratory distress and increased breathing load [42]. The (a)synchrony mechanism depends on the heavy-handed activity of the respiratory muscles, such as inspiratory accessory muscles and the diaphragm, which bring forth changes in lung volume [43]. On the other hand, hyperventilation or over-breathing occurs when there is a drop in arterial carbon dioxide, exceeding metabolic demands of oxygen due to ventilation, which in turn causes a rise in blood pH, that is, respiratory alkalosis [26]. Additionally, hyperventilation is usually described by a shift from a diaphragmatic to a thoracic breathing pattern, which imposes biomechanical musculoskeletal stress on the neck/shoulder region, namely in the muscles that support thoracic breathing [44].

Several parameters have shown valuable use in discriminating healthy respiratory patterns from those with neuromuscular diseases, such as phase angle, percent of rib cage contribution to breathing, respiratory rate, and labored breathing index [45]. Therefore, it was demonstrated that the relationship between respiratory function and fatigability can be explored as an occupational risk indicator. Nevertheless, it remains unclear if internal respiratory synchrony is altered due to fatigue. Furthermore, there is a lack of literature on evaluating the feasibility of using respiration to monitor repetitive work from industrial operators. On the other hand, it has been shown that physiological and subjective fatigue outcomes are not always consistently correlated due to the large variability of requirements from different work settings [46]. Accordingly, this study aims to verify thoraco-abdominal (a)synchrony relationship with fatigability while performing simulated repetitive work. In addition, this work also analyzes the relationship between thoraco-abdominal (a)synchrony and local muscle load.

The rest of this work is organized as follows: Section 2 outlines the study methods; the results are presented in Section 3 and discussed in Section 4; finally, Section 5 summarizes the contributions of this work, along with some recommendations for future work.

## 2. Materials and Methods

To quantify the problem under study, i.e., the susceptibility of using the synchronization between chest and abdominal wall movements as an indicator of fatigability/fatigue, respiratory inductance plethysmography (RIP) was used. To supplement this assessment with local fatigue and alterations that can occur in the movement of the participants, EMG and accelerometry data were also collected. This section begins by presenting the characterization of the voluntaries that participated in this study. Then, the task that was used to simulate repetitive work is explained in detail. Next, the instrumentation of the participants, equipment used, and steps in processing the different signals are provided. Finally, the statistical analysis used in the study is described.

### 2.1. Participants

A total of 22 healthy participants (11 females and 11 males; age: 27.0 ± 8.3; height: 1.72 ± 0.09 m; mass: 63.4 ± 12.9 kg) were recruited for this study. The protocol was clearly explained, and all subjects gave their informed consent for inclusion before they participated in the study. The study was conducted in accordance with the Declaration of Helsinki, and the protocol was approved by the Ethics Committee of University of Porto.

### 2.2. Task

The protocol included three trials of 10 min duration each (Baseline, Fatigue 1, and Fatigue 2). Between trials, a fatigue protocol was performed to ensure a loss of muscle force during the second and third tasks. Data from two minutes of rest before and after the three 10-min trials were also acquired. The repetitive task performed during each 10-min trial was chosen to contain basic motions required to perform work, such as reaching, grasping, releasing, moving, and positioning an object. The task starts from the neutral pose in the upright position, grabbing a box from a lower shelf of a locker. Then, the participant moves the box, placing it on the upper shelf. Immediately after returning to the neutral pose, the participant grabbed the box on the upper shelf, moving it to the lower shelf (see Figure 1). This description corresponds to one cycle with an ascending phase (moving the box from the lower shelf to the upper one) and a descending phase (moving the box from the upper shelf to the lower one). The periodicity among cycles was self-paced. The shelves’ height was adjusted according to the subject’s morphology: the lower shelf was placed at the height of the medium finger while the participant was standing in the upright position; the upper shelf was adjusted so that the box was placed at the eyes level. The box was set at 5% of the participant’s mass.

The fatigue protocol comprised three trials of 10–15 repetitions each until achieving muscular failure. The resistance was set at 65–75% of maximal force, starting with an estimated load according to participants body mass. To ensure that muscle failure occurred in the interval of 10 to 15 repetitions, the following procedure was performed: when the participant reached muscle failure before being able to complete 10 repetitions, the load was decreased according to the number of repetitions performed, and this attempt had to be repeated; when the participant exceeded 15 repetitions, the load was increased so that muscle failure could occur in the mentioned interval, repeating the attempt. During the last repetition of each trial, the participants were encouraged to perform one last repetition with the researcher’s help.

### 2.3. Respiratory Inductance Plethysmography

RIP is a non-invasive method to assess lung volume using elastic bands around the thorax and abdomen [45,47]. Transducers (Plux, Lisbon, Portugal) were placed at the rib cage (at the level of the nipples in males, and over the chest passing underarms in females) and at the umbilicus to monitor changes in trunk volume during inspiration and expiration cycles [48].

The raw signal was sampled at a rate of 1000 Hz (this sample rate was used due to EMG synchronization) and Finite Impulse Response bandpass filtered between 0.15 and 0.45 Hz [49]. The signal was then downsampled to 40 Hz and submitted to Masked Sift Empirical Mode Decomposition [50,51,52] to extract the first intrinsic mode function. Finally, the signal was z-normalized and a smoothing procedure with a sliding average of 10 s was applied. Rolling window correlation and instantaneous phase synchrony between chest and abdominal respiration signals were computed.

The instantaneous phase synchrony measures the similarities regarding the phase (angles) between two signals at each point. Its calculation includes the following steps: (i) computation of discrete Hilbert transform of each signal (summarized in Equation (1)) [53], (ii) extraction of the phase of each signal (imaginary part of the transform, represented in Equation (2)), and (iii) calculation of the difference between the two phases (Equation (3)). In Figure 2, three different examples of respiration signals are presented: the first where the subject is in a resting state, the second during the first trial and the third and last during the last trial.
(1)$$g(n)=\{\begin{array}{ll} \frac{2}{\pi }\;\sum_{t\;odd}\frac{f(t)}{n-t} & n\;even\\ \frac{2}{\pi }\;\sum_{t\;even}\frac{f(t)}{n-t} & n\;odd \end{array}$$
(2)∅=imag(g(n))
(3)$$\varphi \left(n\right)=1-\sin\left(abs\left(\emptyset_{chest}\left(n\right)-\emptyset_{abdominal}\left(n\right)/2\right)\right)$$

### 2.4. EMG

Surface EMG system (Plux, Lisbon, Portugal) with disposable adhesive Ag/AgCl electrodes were used to acquire bilaterally muscle activity from the upper trapezius, anterior deltoideus, biceps brachii, and triceps brachii. Data collection was performed using the acquisition software OpenSignals (Plux, Lisbon, Portugal) with a sample rate of 1000 Hz. Hair removal, abrasion, and alcohol cleaning were performed prior to electrode placement to improve skin-electrode conductivity. The participant was asked to contract the muscles of interest prior to fixation of the electrodes to help place the electrode over the muscle belly. The electrodes were placed at a 20 mm center-to-center distance parallel to the muscle fibers according to Seniam Recommendations [54]. Trapezius’ electrodes were placed at 50% (bulkier area of the muscle) on the line from the acromion to the spine on vertebra C7 as the reference; deltoideus anterior electrodes were set up at one finger width distal and anterior to the acromion with the orientation of the line between the acromion and the thumb; the electrodes on the biceps brachii were placed at 1/3 from the fossa cubit between this localization and the medial acromion. We only considered the dominant limb muscles. The reference electrode was placed on the olecranon surface.

The raw data were stored and digitally filtered with a 4th order bandpass Butterworth filter with cutoff frequencies of 10 and 500 Hz. Then, the root mean square was applied to the signal. The median frequency was used to quantify the fatigue in terms of frequency-domain with its power spectral density of the raw EMG, using Welch’s modified periodogram. Time-domain analysis comprised the normalization process based on MVC. The smoothed signals were normalized by the mean value of the 100 ms window around the maximum value acquired from MVC.

MVC procedure was performed in standing position for both muscles. Each muscle was submitted to three trials of 3–4 s with a rest interval of 20 s between them. Before starting to collect, participants were instructed about each MVC exercise position and how to produce force. The participants began producing force gradually against researcher’s resistance to avoid EMG bursts. For the biceps brachii MVC, the participants’ elbows were ipsilaterally placed at 90° with arms parallel and leaning against the trunk. The force production was exerted against the researcher’s hands that offered resistance in the region of the participant’s wrists. For trapezius’ MVC, the participant shrugged his shoulders as much as he could against the resistance of the investigator’s hands. During this exercise, the researcher was positioned posteriorly in relation to the participant. Verbal encouragement was used to motivate participants to attain maximum performance during each MVC exercise.

### 2.5. Accelerometer

A triaxial accelerometer (Plux, Lisbon, Portugal) was placed on the right wrist of each participant. Accelerometer XYZ axes were digitally filtered with a 4th order bandpass Butterworth filter with cutoff frequencies of 0.1 and 10 Hz. Then, each signal was smoothed with an average window of 0.2 s. The *y*-axis was used to establish the task cycles within each trial (baseline, fatigue 1, and fatigue 2). For this purpose, Hilbert transform was applied, and the normalized transition maximum was used to determine the start and the end of each cycle. Then, visual inspection of each trial was performed to ensure the adequate cycle segmentation. To quantify wrist motion, accelerometer axes were used to calculate the pitch and the roll through the following Euler equations:
(4)$$\mathrm{Pitch}=\arctan(-\frac{{Acc}_{x}}{\sqrt{{Acc}_{y}^{2}+{Acc}_{z}^{2}}})$$
(5)$$Roll=arctan\left(\frac{Acc_{y}}{Acc_{z}}\right)$$
where $Acc_{x}$, $Acc_{y}$, and $Acc_{z}$ correspond to the *xyz* axes acceleration.

Having as reference the moment when the participant is grabbing the box, the *x*-axis, *y*-axis, and the *z*-axis correspond to the medio-lateral, antero-posterior, and superior-inferior directions, respectively. Thus, the pitch can be essentially associated with the pronation/supination of the forearm and adduction/abduction of the arm. The roll can be related to flexion/extension of the forearm and arm.

### 2.6. Statistical Analysis

The statistical analysis comprises mixed effects regression modeling considering inter-participant variability as random effect. For this purpose, the Python package statsmodels was used [55]. The choice of this analysis was due to its greater flexibility in dealing with inter-subject variability over time and between trials in repeated measures designs. Two approaches were followed: (1) one of the dependent variables such as RIP correlation, RIP synchronization, EMG amplitude, EMG median frequency with the following independent variables: trials (Baseline, Fatigue 1, and Fatigue 2) and divisions of each trial (each trial was divided by deciles parts, corresponding about 1 min each); (2) as dependent variables the RIP correlation or RIP synchronization and as independent variables the EMG amplitude and EMG median frequency keeping the trials and the divisions in each trial as previously stated. The assumptions of independence, normality, and homoscedasticity of residuals were verified using the Wald test, Shapiro–Wilk test, and the White’s Lagrange Multiplier test, respectively. Product moment of Pearson was applied to quantify the correlation between significant muscle activity contribution (in the mixed effects model) and RIP correlation and/or synchrony. Normality correction was performed using Yeo-Johnson power transformation [56]. After this transformation, z-score was applied to avoid mean and standard deviation range bias among variables due to the previous power normalization.

Regarding accelerometer results, that is, to verify whether the number of cycles, pitch, and roll change over the trials, a repeated measures ANOVA was used. The sphericity was verified through the Mauchly’s test, and when it was not present the Greenhouse-Geisser correction was performed. Pairwise comparisons were performed with paired t-tests with Bonferroni correction. This analysis was done with the Python package pingouin [57]. The level of significance was set at 5%.

## 3. Results

This section begins by presenting the results of RIP correlation and phase synchronization and muscle activity separately. Then, the relationship of muscle activity with both RIP correlation and phase synchronization is explored. Finally, we present accelerometry results that account for changes in movement across the three trials. The figures presented for RIP correlation and phase synchronization, EMG amplitude, and median EMG frequency show real values. In contrast, the statistical results presented in the tables stem from the transformed values to ensure the applicability assumptions.

### 3.1. RIP

Figure 3 and Figure 4 illustrate the mean and standard deviation of rib cage and abdominal wall correlation and phase synchronization for each trial (Baseline, Fatigue 1, and Fatigue 2) across their deciles. Table 1 shows the mixed effect models results taking the rib cage and abdominal correlation and phase synchronization as dependent variables.

The mean value of correlation was 0.65 ± 0.20 for Baseline, 0.57 ± 0.21 for Fatigue 1, and 0.58 ± 0.20 for Fatigue 2. Phase synchrony showed coefficients of 0.65 ± 0.10, 0.61 ± 0.11, and 0.62 ± 0.09 for Baseline, Fatigue 1, and Fatigue 2, respectively.

Fatigue 1 and Fatigue 2 presented significant differences from baseline condition in both RIP (*p* < 0.001) correlation and phase synchronization (*p* < 0.001). The division only showed significant differences for RIP synchronization in decile 10 (*p* = 0.026) for the correlation.

### 3.2. EMG

Similar to the presentation for RIP results, Figure 5 and Figure 6 show the evolution of biceps and trapezius EMG amplitude using %MVC and median frequency, during each trial across their deciles, respectively. The biceps brachii was activated at 16.7 ± 5.9% MVC during Baseline, increasing to 19.0 ± 6.4% MVC in the Fatigue 1 trial, and then, achieving 18.2 ± 5.8% MVC in Fatigue 2, showing a significant main effect (*p* < 0.001). The median frequency was about 62.8 ± 8.4 HZ, 63.3 ± 8.3 HZ, and 62.2 ± 8.5 HZ, but the biceps only showed a significant effect in Fatigue 2 trial (*p* = 0.007) and was similar among trials (*p* > 0.050).

Trapezius muscle also increased significantly from trial to trial (*p* < 0.001), showing amplitude values of 12.7 ± 5.3% MVC, X ± Y 13.5% MVC, and 14.2 ± 6.3% MVC. Median frequency also presented a significant effect, but contrary to biceps brachii, it was present in both Fatigue trials (*p* < 0.010), presenting values of 78.5 ± 11.9 Hz, 80.8 ± 12.9, and 79.7 ± 13.1 Hz across the three trials. The statistical results can be seen in Table 2.

### 3.3. RIP vs. EMG

The following analysis was performed to verify whether the muscle activity has a significant effect on RIP correlation and phase synchronization despite having revealed significant differences across trials. As in previous models, there were significant main effects of the trials predicting RIP correlation and phase synchrony for both amplitude and frequency analysis *p* ∈ [<0.001, 0.003]); however, only trapezius activation showed a significant effect *p* ∈ [<0.001, 0.001]) in the prediction of these two parameters regarding amplitude of activation. On the contrary, only the biceps brachii had a significant association of the median frequency (*p* = 0.006), but just for phase synchrony prediction, not for correlation (*p* = 0.115). The statistical results can be seen in Table 3.

Figure 7 illustrates joint plots with the distribution and scatter plot of the muscle activity variables that significantly contributed to RIP correlation and phase synchronization in the previous mixed-effect models. Of these variables, a small and negative correlation was found between trapezius amplitude (average EMG) and both RIP correlation (r=−0.282, p<0.001) and phase synchrony (r=−0.295, p<0.001) and positive between and biceps median frequency and phase synchrony (r=−0.241, p<0.001).

### 3.4. Accelerometer

Figure 8 illustrates the results for the number of cycles performed by the participants and both pitch and roll of the accelerometer placed on the wrist. No significant differences were found for the number of cycles ($F_{\left(1.436,\;30.164\right)}=1.883;p=0.165;\;\eta_{p}^{2}=0.082$) despite the decreasing tendency.

Significant differences were found for the pitch ($F_{\left(2,\;42\right)}=11.537;p<0.001;\;\eta_{p}^{2}=0.355$) between the baseline and the two fatigue trials (*p* = [0.001, 0.013]). The roll showed significant differences ($F_{\left(2,\;42\right)}=4.403;p=0.018;\;\eta_{p}^{2}=0.173$) between the two fatigue trials (p=0.041), and it was tendencialy between the baseline and the Fatigue 2 trial (p=0.050).

## 4. Discussion

This study analyzes the evolution of RIP correlation and synchronization as fatigability increases during a simulated protocol of repetitive work. Accordingly, the RIP relationship with EMG (amplitude and median frequency) was also studied to analyze the association between respiration adjustments with local fatigue. To achieve these purposes, participants were asked to perform three trials of 10 min each while performing a repetitive task. Between each pair of trials, a fatigue protocol was performed to ensure that an agonist muscle will act after muscular failure. The participant performed the task after local fatigue which should influence the general body adjustments while, repeatedly, he/she is trying to accomplish the task fighting against tiredness. RIP correlation and phase synchronization represent a response to these adjustments. Our results showed that RIP correlation and phase synchronization change from baseline when compared to fatigue trials, showing tendentially a decrease in thoracic-abdominal synchronization across trials. These results were in agreement with the local fatigability of the agonist muscle (i.e., biceps brachii) and the shoulder fixator (i.e., upper trapezius) by means of EMG amplitude rather than median frequency. However, only the trapezius activation showed a significant association with both RIP correlation and phase synchrony. Although the median frequency of biceps brachii had a significant association with phase synchrony, the results were not consistent with RIP correlation. In addition, there was found inter-subject variability suggesting different motor strategies.

### 4.1. RIP

The thoracic-abdominal synchronization is dependent on the coordination of how respiratory muscles act as oscillators during respiratory cycles. The efficiency of this action is correlated with their inspiratory muscle strength [58]. Inspiratory muscle strength will be limited if the driving pressure is reduced or when vascular conductance reaches its maximum capacity regarding the requirements, resulting in some kind of alveolar hypoventilation and hypercapnia [59]. In case of fatigue or as the demand is increasing, the central nervous system through the respiratory centers in the brainstem stimulate the inspiratory neurons to increase ventilation in order to promote oxygen supply and remove carbon dioxide, maintaining the acid-basic balance. Thus, as the contraction requirement of non-respiratory muscles increases, so does the contribution of respiratory muscles. Although this RIP technique has limitations quantifying respiratory muscle fatigue [42], a strong relationship between physical demand and respiratory response was reported while performing construction tasks [21]. Thoracic dysfunctional breathing can be provoked by both physiological or psychological stress, in which biomechanical patterns may be respiratory and/or non-respiratory [60]. Thus, our article explores the relationship of thoracic-abdominal synchronization when non-respiratory muscles are susceptible to fatigability during repetitive tasks. Commonly, the variables that are used to quantify the thoracic-abdominal asynchrony are phase difference (φ), percentage of rib cage contribution, and labored breathing index (e.g., [61,62]).

Respiratory muscles act as mechanical effectors and can be divided into three main groups: (1) the inspiratory muscles, (2) the expiratory muscles, and (3) the accessory muscles of respiration [63]. The inspiratory muscle that has the greatest preponderance in inspiration is the diaphragm, whose contraction causes it to descend, increasing both abdominal pressure and thoracic volume. The two sheets of intercostal muscles, the internal and external intercostal muscles, contribute to expanding the rib cage during inspiration and to decreasing rib cage volume during expiration, respectively. The abdominal muscles mainly participate in expiration but also contribute to inspiration, having a relevant role in storing elastic recoil energy in the chest wall, assisting its spread for the next inspiration [63].

Respiratory muscle failure due to fatigue or disease results in a reduction of the capacity in generating force when these muscles’ energy supply is not enough during increased work demand, or when respiratory work requirements are exceeded [64,65]. In this scenario, the fine coordination between inspiratory and expiratory muscles is compromised in the role of avoiding rib cage distortion and lung pressure control, increasing thoraco-abdominal asynchrony [66]. In our study, the phase difference was the chosen metric complemented by the signal correlation. The highest phase asynchrony should occur when one compartment moves in the opposite direction to the other.

Our results showed synchrony alterations when fatigue is present, namely after action with higher local muscle requirements. Both RIP synchronization and correlation decrease when comparing baseline to fatigue trials. Regarding continuous working without stopping, only RIP correlation was sensitive to changes between rib cage and abdominal walls, showing a significant association during the 7th and 10th decile. This disrupted alteration across the trial could represent motor control adjustments due to proprioceptive afferent feedback.

Usually, two major types of fatigue are reported (e.g., [67,68]): central and peripheral. Central fatigue occurs from the synthesis and metabolism of central monoamines impairment, resulting in a reduction in motor unit recruitment, which, in turn, could negatively alter physical and mental efficiency [69]. Peripheral fatigue occurs either due to disabled impulse propagation across the neuromuscular junction and/or over the muscle surface membrane or due to the cross-bridges cycle impairment [65], which biomarkers depend on the type of effort and underlying muscle energy mechanism [70].

In the present study, two EMG methods were used to quantify muscle fatigue or fatigability: amplitude normalized from MVC and median frequency. It is expected that, as fatigue increases, the amplitude of the signal also increases, and the opposite occurs with the median frequency due to the shift toward lower frequencies in the power spectrum. In our study, the EMG amplitude showed higher sensitivity than median frequency regarding repetitive work. Although there is a decrease during each 10-min trial, the median frequency change was not significant among trials. One can point out that the shift of median frequency to lower frequencies depends on alterations of motor units’ potentials actions, showing an increase in their duration and a decrease in the conduction velocity [71]. However, we studied a dynamic rather than isometric that was performed at an intensity lower than 30% of MVC [72], compromising the visible shift of frequencies to lower parts of the frequency spectrum. The task of our study was designed to mimic (as much as possible) repetitive work that was reported to be lower than 20% of MVC [24].

The mean activity of the trapezius in our study was lower than 20% of MVC, but its amplitude showed significant effects on both RIP correlation and phase synchronization rather than biceps brachii. Interestingly, other studies have reported the fatigue of this muscle during repetitive work or long periods of standing [73,74]. This occurs due to the role of the shoulder allowing kinematic and muscular adjustments to promote performance sequentially during prolonged repetitive work [74]. On the other hand, the biceps brachii achieves higher activation in the second trial compared to the third, which suggests biomechanical adaptations due to localized fatigue that are compensated with other muscles or motor strategies, as previously demonstrated during a cyclic lifting task [75]. Indeed, the accelerometer placed on the wrist (Supplemental Material) showed from trial to trial a decrease in the anterior-posterior movement and an increase of medio-lateral. This could mean that as fatigability increases in the agonist muscle (the biceps brachii in this case) the main action of flexion/extension starts to become compromised leading to more abduction/adduction, and then higher activation of the shoulder complex muscles. This type of behavior is according to others that reported a repositioning of joints to maintain task performance during repetitive work [74,76].

### 4.2. RIP vs. Muscle Activity

To analyze the contribution of local fatigue with alterations in RIP synchronization and phase synchrony, mixed effect regression modeling was performed, adding muscle activity as independent variables. In EMG amplitude, only trapezius muscle showed a significant association with both RIP correlation (*p* < 0.001) and phase synchrony (*p* = 0.001). However, only the biceps brachii presented a significant contribution (*p* = 0.005) to the model of phase synchronization; but not for correlation (*p* = 0.126). These results highlight the role of trapezius in the shoulder to promote joint repositioning and compensatory action due to fatigue of agonist muscles; as stated in the previous paragraph. Accordingly, the muscles’ local fatigue produces changes in the general functional profile as a consequence of adaptations of motor control strategy [77]. Joint movement trade-offs to redistribute loads across the upper extremity and trunk during repetitive work were previously reported [1], allowing to continue task performance with the presence of fatigue.

The joint repositioning and compensatory alterations due to the local fatigue of the studied agonist muscle and the higher activation of trapezius are supported by the accelerometry results, although limited to the wrist information only. Both pitch and roll tend to increase in amplitude over the trials. In the case of pitch, the baseline differs from both fatigue trials, but not when fatigue is already installed (i.e., between fatigue 1 and fatigue 2 trials). In the case of roll, it is the last fatigue trial (i.e., fatigue 2) that differs from both the baseline and the previous fatigue state. This occurs without significant differences in the number of cycles over the three trials, which in turn reinforces that to maintain performance during accumulated fatigue, the participants performed adaptations of motor control strategy, as stated before [77]. On the one hand, the participants tend to increase pronation/supination of the forearm and/or adduction/abduction of the shoulder during fatigue conditions to compensate for the wear and tear of agonist muscle activity. On the other hand, to maintain performance when accumulated fatigue is present, paradoxically, it increases again the flexion/extension of the shoulder and elbow.

Additionally, due to non-ventilatory functions in tasks where arms are unsupported, some of the inspiratory muscles of the rib cage decrease their ventilatory contribution, shifting the dynamic work to the diaphragm and abdominal muscles of exhalation [78], which leads to RIP asynchrony. Moreover, the task used in this study was performed in the standing position requiring permanent re-establishment adaptation to keep balance while grabbing, placing, and moving a box concentrically and eccentrically.

Another issue that should be taken into consideration is that these motor strategies are not so easy to standardize because there is a high inter-subject variability [1]. We also encountered this issue in the present study, as shown in Figure 7. All the participants changed their behavior through the studied variables, but not in the same fashion.

### 4.3. Limitations

We highlight two limitations: (1) Statistical analysis was performed using data transformation to guarantee the assumptions of applicability due to inter-subject variability, instead of raw values; (2) a limitation that can compromise the reproducibility of studies is the determination of the respiratory cycle in dynamic actions, since the signal becomes contaminated with adaptations of the thoracic and abdominal grid wall due to the movement of the upper limbs. The expansion of the trunk volume is not merely dependent on the influence of the respiratory cycles, but also on the action of other body segments. For example, the action of flexion/extension and abduction/adduction movements of the arms will be incorporated into the respiratory signal of chest volume expansion/reduction. Unlike at rest, the disparity of different actions in labor context and the adaptation of the necessary processing may lead to divergent interpretations.

### 4.4. Future Work

The choice of the data collection setup with multiple sensors used in our study was due to the need to understand these indicators in a controlled environment such as a laboratory. Thus, the main recommendation for future work is to collect this type of data in a real environment setting, searching for changes as the working shift lasts. Likewise, comparing different types of workstations and workers’ morphological conditions is recommended. It should be relevant to study RIP correlation and synchronization in both between and within designs during different shifts and chronotypes since both sleep [79] and circadian cycles [80] were associated with work-related fatigue. Studies that comprise inter-subject variability also should be considered to better adjust intervention to individual motor strategies. A great challenge will be to establish the RIP guidelines in which dropping below a certain value is advisable for workers to take a break, namely considering the individual variability.

## 5. Conclusions

Respiratory inductance plethysmography correlation and synchronization during repetitive work change after muscle failure of tasks’ agonist muscle. Accordingly, both the agonist (biceps brachii in this case) and the shoulder stabilizer muscle (trapezius) increase their amplitude activity from Baseline to Fatigue status. The median frequency of the latter was more discriminative in Fatigue trials than biceps brachii, which in turn showed significant effects only during the second Fatigue task. Remarkably, the biceps brachii amplitude showed significant effects across the 10-min trials, namely from the middle to the end of the trial duration.

Despite the previous results for Bicep brachii amplitude, this muscle did not show a significant effect on RIP correlation and synchronization, unlike the trapezius muscle. This muscle showed a significant contribution to predicting both respiratory parameters. This result could be linked to compensatory activity of trapezius in repositioning the shoulder complex to maintain task performance during repetitive work.

This study supports that RIP can be an interesting quantitative approach to evaluate fatigability during repetitive work because it has been shown that this tool is sensitive to fatigue, is easy to use, and is a non-invasive solution. Thus, it might help decision-makers in ergonomic risk assessment studies and interventions as well as in tasks distribution throughout the day in the production line.

## Figures and Tables

**Figure 1 sensors-22-04247-f001:**
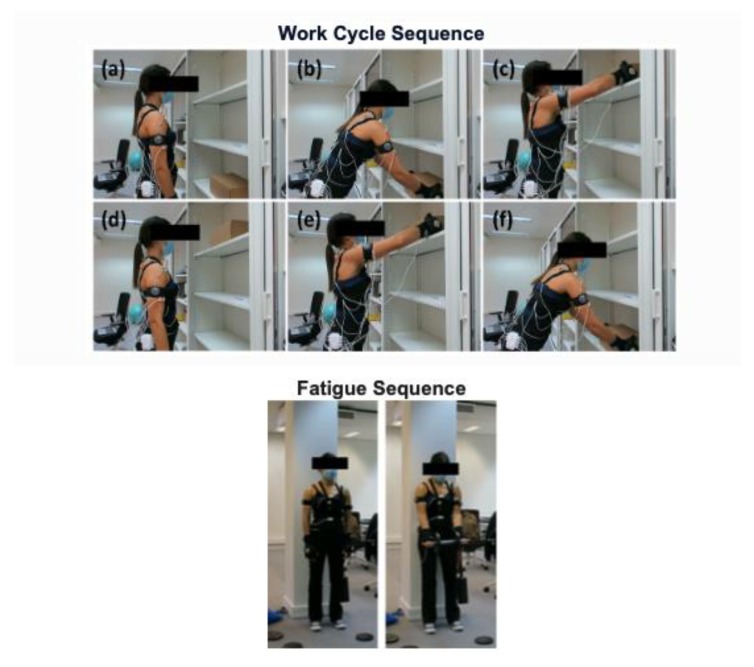
Work cycle sequence: (**a**) Neutral position at the beginning of the cycle; (**b**) grabbing the box from the lower shelf; (**c**) placing the box on the upper shelf; (**d**) neutral position after returning from the upper shelf placement; (**e**) grabbing the box from the upper shelf; (**f**) placing the box on the lower shelf. Fatigue sequence: three trials of biceps curl until muscular failure in each one.

**Figure 2 sensors-22-04247-f002:**
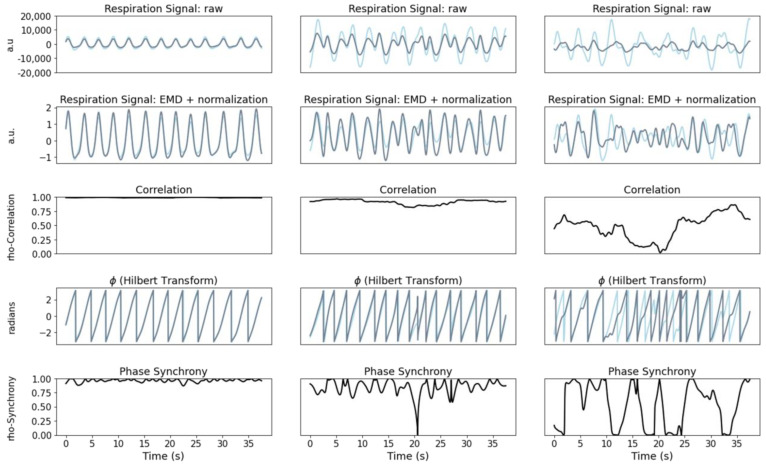
Example of a participant during rest (left column), baseline (middle column), and after fatigue protocol (right column): raw respiration signals (first row), respiration signals after EMD and normalization (second row), and signal phase (fourth row). In each of these figure rows, the chest movement (light blue) and abdominal movement (dark blue) walls are represented. Correlation and phase synchrony are shown in the third and fifth rows, respectively (both in black).

**Figure 3 sensors-22-04247-f003:**
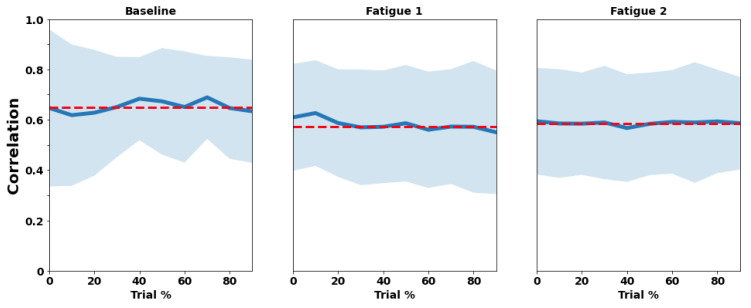
Trial mean (red dashed line), means of each decile (blue line), and standard deviation (light blue shadow) of rib cage and abdominal walls correlation for each trial (Baseline, Fatigue 1, and Fatigue 2) across their deciles (trial %). The values of correlation range from “0” to “1”, where 0 corresponds to no correlation and “1” to total correlation.

**Figure 4 sensors-22-04247-f004:**
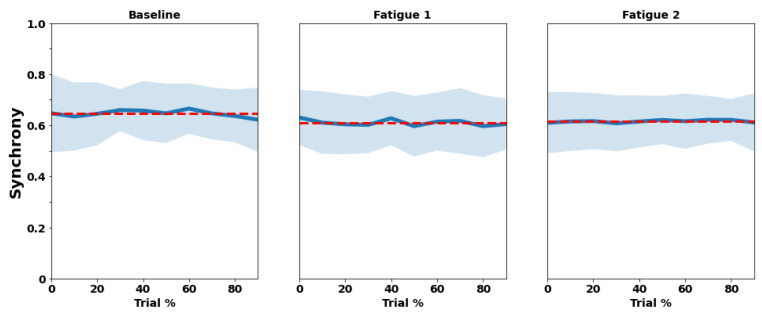
Trial mean (red dashed line), means of each decile (blue line), and standard deviation (light blue shadow) of rib cage and abdominal walls phase synchronization for each trial (Baseline, Fatigue 1, and Fatigue 2) across their deciles (trial %). The values of correlation range from “0” to “1”, where 0 corresponds to no synchrony at all (antiphase) and “1” to total synchronization (in phase).

**Figure 5 sensors-22-04247-f005:**
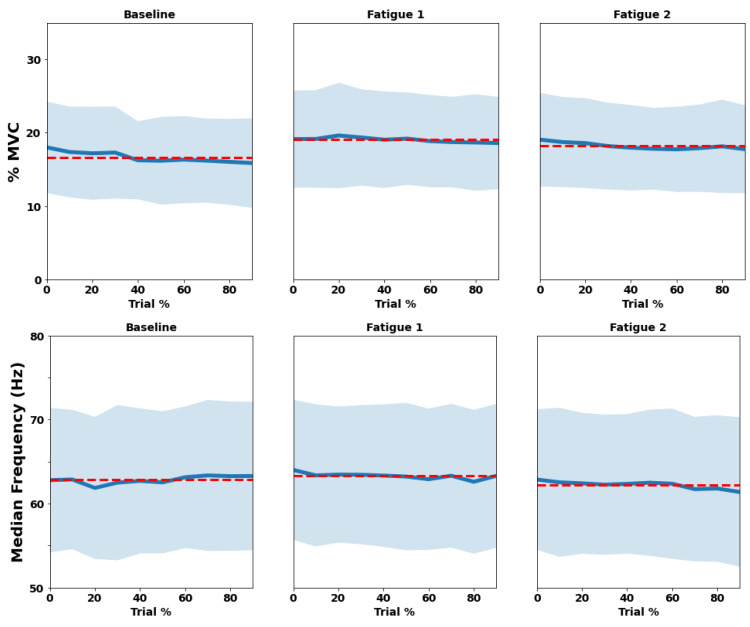
Trial mean (red dashed line), means of each decile (blue line), and standard deviation (light blue shadow) of both amplitude (upper) and median frequency (bottom) among the three trials (baseline, Fatigue 1, and Fatigue 2) for biceps brachii, considering their deciles (trial %).

**Figure 6 sensors-22-04247-f006:**
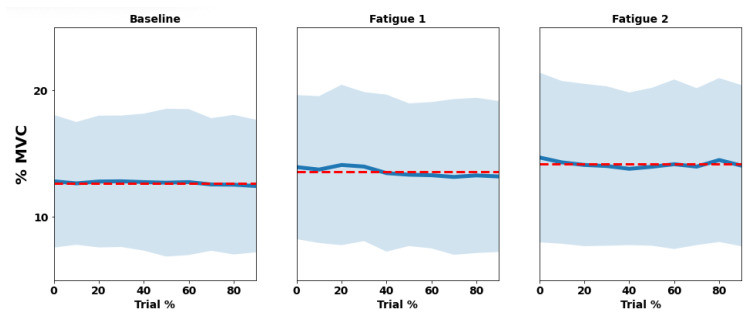
Trial mean (red dashed line), Means of each decile (blue line), and standard deviation (light blue shadow) of both amplitude (upper) and median frequency (bottom) among the three trials (baseline, fatigue 1, and fatigue 2) for trapezius, considering their deciles (trial %).

**Figure 7 sensors-22-04247-f007:**
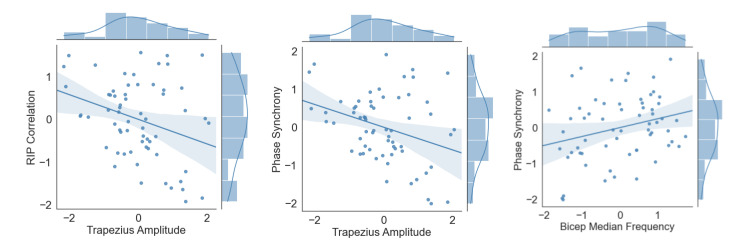
Relationship between RIP correlation/phase synchrony and muscle activity that have shown significant effects in the mixed-effects model. The arbitrary units are due to the Yeo-Johnson power transformation.

**Figure 8 sensors-22-04247-f008:**
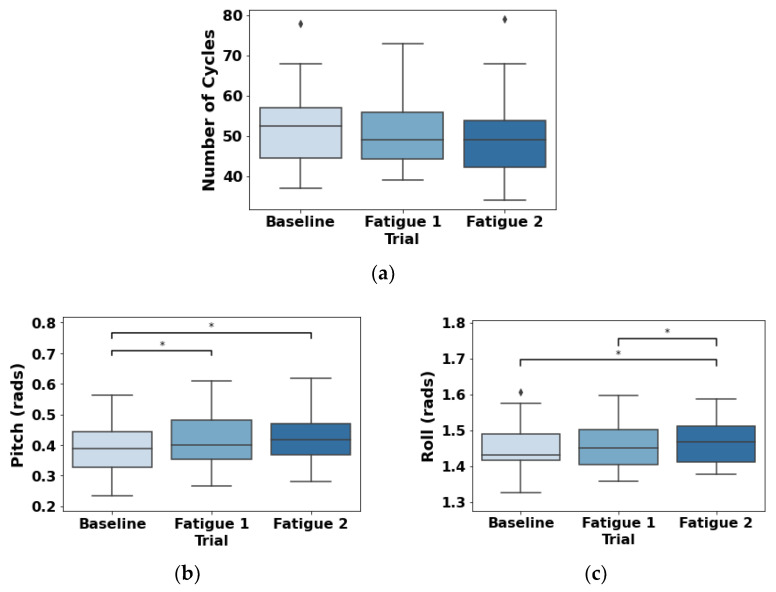
Number of cycles (**a**), pitch (**b**), and roll (**c**) results for the three trials (baseline, Fatigue 1, and Fatigue 2). * means significant differences.

**Table 1 sensors-22-04247-t001:** Mixed effect regression with the rib cage and abdominal walls correlation and synchronization as dependent variable and trials and divisions as independent variables.

		RIP Correlation			RIP Synchrony		
		β	*Z*	*p*	β	*Z*	*p*
	Intercept	0.373	1.928	0.054	0.275	1.414	0.157
**Trial**	Fatigue 1	−0.364	−6.153	<**0.001**	−0.324	−5.490	<**0.001**
	Fatigue 2	−0.348	−5.881	<**0.001**	−0.290	−4.904	<**0.001**
**Division**	2	−0.095	−0.878	0.380	−0.082	−0.758	0.449
	3	−0.169	−1.563	0.118	−0.041	−0.380	0.704
	4	−0.130	−1.207	0.228	−0.068	−0.635	0.525
	5	−0.138	−1.278	0.201	0.018	0.163	0.870
	6	−0.095	−0.882	0.378	−0.128	−1.183	0.237
	7	−0.207	−1.910	0.056	−0.041	−0.376	0.707
	8	−0.120	−1.112	0.266	−0.049	−0.453	0.650
	9	−0.153	−1.418	0.156	−0.133	−1.233	0.217
	10	−0.241	−2.23	**0.026**	−0.179	−1.658	0.097
**Subject**	Variation	0.637			0.647		

Legend: significant values are in bold.

**Table 2 sensors-22-04247-t002:** Mixed effect regression with biceps brachii and trapezius EMG amplitude and median frequency as dependent variables.

		Amplitude						Frequency					
		Biceps Brachii			Trapezius			Biceps Brachii			Trapezius		
		β	*Z*	*p*	β	*Z*	*p*	β	*Z*	*p*	β	*Z*	*p*
	Intercept	−0.096	−0.467	0.641	−0.066	−0.304	0.761	0.076	0.352	0.725	−0.061	−0.278	0.781
**Trial**	Fatigue 1	0.433	10.207	**<0.001**	0.141	5.727	**<0.001**	0.050	1.423	0.155	0.169	6.966	**<0.001**
	Fatigue 2	0.294	6.93	**<0.001**	0.245	9.985	**<0.001**	−0.094	−2.697	**0.007**	0.071	2.913	**0.004**
**Division**	2	−0.058	−0.753	0.451	−0.039	−0.879	0.379	−0.046	−0.72	0.472	0.017	0.378	0.705
	3	−0.060	−0.776	0.438	−0.030	−0.671	0.502	−0.091	−1.427	0.154	−0.035	−0.785	0.432
	4	−0.089	−1.15	0.250	−0.036	−0.798	0.425	−0.061	−0.965	0.335	−0.052	−1.182	0.237
	5	−0.189	−2.434	**0.015**	−0.078	−1.745	0.081	−0.068	−1.075	0.282	−0.03	−0.669	0.503
	6	−0.195	−2.515	**0.012**	−0.086	−1.912	0.056	−0.062	−0.968	0.333	−0.042	−0.947	0.343
	7	−0.200	−2.576	**0.010**	−0.078	−1.735	0.083	−0.058	−0.912	0.362	−0.032	−0.71	0.478
	8	−0.195	−2.517	**0.012**	−0.094	−2.102	**0.036**	−0.059	−0.922	0.356	−0.033	−0.747	0.455
	9	−0.226	−2.913	**0.004**	−0.073	−1.624	0.104	−0.092	−1.44	0.150	0.020	0.448	0.654
	10	−0.251	−3.247	**0.001**	−0.109	−2.428	**0.015**	−0.071	−1.118	0.263	−0.005	−0.108	0.914
**Subject**	Variation	0.814			0.973			0.914			0.980		

Legend: significant values are in bold.

**Table 3 sensors-22-04247-t003:** Mixed effect regression with the rib cage and abdominal walls correlation and synchronization as dependent variables adding muscle activity to the models.

		Amplitude						Frequency					
		RIP Correlation			RIP Synchronization			RIP Correlation			RIP Synchronization		
		β	*Z*	*p*	β	*Z*	*p*	β	*Z*	*p*	β	*Z*	*p*
	Intercept	0.342	1.818	0.069	0.245	1.304	0.192	0.355	1.825	0.068	0.249	1.288	0.198
**Trial**	Fatigue 1	−0.283	−4.503	**<0.001**	−0.238	−3.79	**<0.001**	−0.345	−5.649	**<0.001**	−0.304	−5.004	**<0.001**
	Fatigue 2	−0.241	−3.863	**<0.001**	−0.185	−2.974	**0.003**	−0.327	−5.438	**<0.001**	−0.259	−4.328	**<0.001**
**Division**	2	−0.113	−1.061	0.289	−0.100	−0.937	0.349	−0.087	−0.808	0.419	−0.07	−0.651	0.515
	3	−0.184	−1.728	0.084	−0.056	−0.528	0.598	−0.164	−1.516	0.130	−0.03	−0.275	0.783
	4	−0.15	−1.404	0.16	−0.088	−0.831	0.406	−0.131	−1.215	0.225	−0.066	−0.614	0.539
	5	−0.18	−1.679	0.093	−0.025	−0.235	0.814	−0.135	−1.248	0.212	0.026	0.238	0.812
	6	−0.14	−1.306	0.192	−0.173	−1.622	0.105	−0.095	−0.875	0.381	−0.123	−1.146	0.252
	7	−0.249	−2.323	**0.020**	−0.084	−0.789	0.43	−0.205	−1.894	0.058	−0.035	−0.326	0.745
	8	−0.168	−1.567	0.117	−0.097	−0.909	0.363	−0.119	−1.098	0.272	−0.043	−0.404	0.686
	9	−0.196	−1.826	0.068	−0.178	−1.663	0.096	−0.14	−1.294	0.196	−0.112	−1.039	0.299
	10	−0.298	−2.774	**0.006**	−0.237	−2.213	**0.027**	−0.234	−2.163	**0.031**	−0.166	−1.544	0.122
**Muscle**	Biceps	−0.076	−1.283	0.199	−0.101	−1.707	0.088	0.112	1.576	0.115	0.194	2.739	**0.006**
	Trapezius	−0.346	−3.709	**<0.001**	−0.304	−3.278	**0.001**	−0.147	−1.530	0.126	−0.178	−1.856	0.063
**Subject**	Variation	0.601			0.599			0.647			0.640		

Legend: significant values are in bold.

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
