# Peer review of "Respiratory Inductance Plethysmography to Assess Fatigability during Repetitive Work"

_sensors, 2022, doi:10.3390/s22114247_

Round 1

Reviewer 1 Report

The authors presented a really interesting and well-designed study on respiratory parameters and its association with fatigue. The methods are well described and results share valuable information to the area.

Still there are some points in the manuscript that need further improvement/clarification.

Methods:

  1. The “Fatigue sequence” was performed between each trial. However, no details were provided on how the fatigue sequence was designed and performed except illustration in Figure 1. How was it ensured that each participant reached fatigue? Was there a subjective rating or any other criteria? Considering the large individual difference, the level of fatigue for each participant in the two “Fatigue trials” could also vary a lot. Was there a subjective rating or index for that?
  2. The MVC procedure for each muscle group should be described.
  3. Statistical analysis:

Line 232 “as independent variables the EMG amplitude of EMG median frequency” should be “the EMG amplitude and EMG median frequency”.

Line 236 “To the significant EMG relationships that were found with RIP correlation 236 and synchrony was applied the product moment of Pearson.” This sentence was hard to understand and should be rephrased.

Results:

  1. The authors showed interesting results of significant differences in the mix effect models (tables 1,2,3). However, the figures (e.g. 4,5,6) were not really showing the significant differences. One possible reason is that there was significant individual variation in the mix effect model. I wonder if the authors should consider another type of plotting to visualize the effect, instead of the current plots showing the average and SD values of the group?
  2. RIP versus EMG, Line 296-305. Clarification e.g. “The statistical results can be seen in Table 3” should be added.
  3. The results from the accelerometers are lacking and should be added. The authors only mentioned in the method and discussion. Is there a difference of the movement between participants? Between the fatigue and baseline trials? Was the pace of tasks decided by each participant? How many lifting was performed during each trial? – These can be relevant information to provide in the manuscript.
  4. Figure 7. I find these plots difficult to understand. The authors should explain them better, and what information do they provide? In addition, Line 447-449, the authors stated “as shown in Figure 7. All the participants 448 changed their behavior through the studied variables”. How was this information provided by Figure 7?

Discussion:

  1. The author talked about central vs. local fatigue in the discussion. In this lab setting, was it aimed to reach local fatigue? Maybe the author can clarify it better in the text.
  2. Line 340-342: “These kinds of measures are critical because physiological and subjective fatigue outcomes are not always consistently correlated due to the large variability of requirements from different work settings [55].” What do the authors mean by this sentence? And how was it supported by the statements before?
  3. Limitation Line 453-456: Can the authors further explain, or discuss more regarding how “the determination of the respiratory cycle in dynamic actions” would compromise the “the reproducibility of studies”? How would it impact the conclusions of this study?

Reviewer 2 Report

The authors present the article entitled ”Respiratory inductance plethysmography to assess fatigability during repetitive work”.

This paper proposes to analyze alterations in respiratory inductance plethysmography (RIP) to measure the asynchrony between thorax and abdomen walls during repetitive and its relationship with local fatigue.

The article presents the following concerns:

  • Lines 44-46 present many coincidences according to the next link: https://www.rssb.co.uk/what-we-do/insights-and-news/blogs/rich-data-from-wearables-improve-our-understanding-of-fatigue rewrite or quote correctly.
  • I recommend giving an introduction in section 2 before participants. 
  • Please add a nomenclature table to define variables and acronyms. 
  • Avoid using apostrophes.
  • The Introduction section is too extensive. I suggest synthesizing this section in order to present a clear background, state of the art review, and the objective of the article by highlighting its contribution. 
  • Line 70: It is not recommended to cite references as follows: (see review: [24]). Also, the Introduction is extensive. Please remark on the main contributions to the proposed work of reference [24].
  • Line 226: Please mention the main advantages of the use of Python package statsmodels and justify the use of these models.
  • Include a table that compares the findings of the work vs the already reported in the stat of the art.
  • line 76 can be justified with these fresher sEMG papers: A study of movement classification of the lower limb based on up to 4-EMG channels; A study of computing zero-crossing methods and an improved proposal for EMG signals; Support vector machine-based EMG signal classification techniques: a Review

Justify the high number of coauthors, preferably 4-5 are fine.

The following misspelling should be checked:

  1. Line 19: “muscle activity are used for cumulative…” should be rewritten as “muscle activity are used for a cumulative”.
  2. Line 30: “synchronization can be used for overall…” should be rewritten as “synchronization can be used for an overall…”
  3. Line 47: “such as job rotation plan…” should be rewritten as “such as a job rotation plan” you were missing the article
  4. Line 48: “do not demonstrate significant effect…” should be rewritten as “o not demonstrate a significant effect” 
  5. Line 114: “ changes in lung volume [42] On the other hand, hyperventilation or over breathing occurs…” should be rewritten as “changes in lung volume [42]. On the other hand, hyperventilation or over-breathing occurs…”
  6. Line 150: “neutral pose in upright position…” should be rewritten as “neutral pose in the upright position…”
  7. Line 152: “the participant grabs the box…” should be rewritten as “the participant grabbed the box…”  grab seems to be in the wrong tense
  8. Line 194: “ electrodes was used to…” should be rewritten as “electrodes were used to acquire“ the verb was does not seem to agree with the subject. change the verb form.
  9. Line 209: “The raw data were stored and digital…” should be rewritten as “The raw data were stored and digitally…”
  10. Line 303: “Biceps brachii had significant association…” should be rewritten as “Biceps brachii had a significant association…”
  11. Line 359: “nonrespiratory” should be rewritten as “non-respiratory” it appears to be missing a hyphen.

Round 2

Reviewer 1 Report

The authors have greatly improved the manuscript accordingly. Only minor comments on the current version that the authors can clarify:

1.     The “Fatigue protocol” was described now in the text. Still, one question remains regarding reaching fatigue. What was judged as fatigue? “If the participant did not reach the 10 repetitions or exceeded 15 repetitions, the trial was performed again. ” How could the participant do it again if he/she could not reach 10 repetitions?

2.     The MVC procedure was added to the text. However, some improvement is needed of the following sentences in order to be understandable: “During biceps brachii contraction, the researcher placed participants' elbow at 90 grabbing comfortably the wrist joint. For trapezius muscle, participants shorten the shoulders as must as possible against researcher's resistance. ” Photos of the procedures can also be added to illustrate what was done.

3.     Analysis on accelerometer result: “Significant differences were found for the pitch…and the roll… between trials”. Seems these parameters all increased its value. Can the authors explain the results, what did the pitch and roll of the wrist-accelerometer show? It would also be helpful to describe or show what directions were the XYZ axes placed on the wrist.

Reviewer 2 Report

My comments have been accomplished 

Author Response

Thank you.